# Surface Functionalisation of Self-Assembled Quantum Dot Microlasers with a DNA Aptamer

**DOI:** 10.3390/ijms241914416

**Published:** 2023-09-22

**Authors:** Bethan K. Charlton, Dillon H. Downie, Isaac Noman, Pedro Urbano Alves, Charlotte J. Eling, Nicolas Laurand

**Affiliations:** Technology & Innovation Centre, Institute of Photonics, University of Strathclyde, 99 George Street, Glasgow G1 1RD, UK; bethan.charlton@strath.ac.uk (B.K.C.); dillon.downie@strath.ac.uk (D.H.D.); isaac.noman@strath.ac.uk (I.N.); pedro.alves@strath.ac.uk (P.U.A.); charlotte.eling@strath.ac.uk (C.J.E.)

**Keywords:** colloidal quantum dots, microsphere lasers, whispering gallery mode laser, surface functionalisation, self-assembled supraparticles, aptamers

## Abstract

The surface functionalisation of self-assembled colloidal quantum dot supraparticle lasers with a thrombin binding aptamer (TBA-15) has been demonstrated. The self-assembly of CdSSe/ZnS alloyed core/shell microsphere-shape CQD supraparticles emitting at 630 nm was carried out using an oil-in-water emulsion technique, yielding microspheres with an oleic acid surface and an average diameter of 7.3 ± 5.3 µm. Surface modification of the microspheres was achieved through a ligand exchange with mercaptopropionic acid and the subsequent attachment of TBA-15 using EDC/NHS coupling, confirmed by zeta potential and Fourier transform IR spectroscopy. Lasing functionality between 627 nm and 635 nm was retained post-functionalisation, with oleic acid- and TBA-coated microspheres exhibiting laser oscillation with thresholds as low as 4.10 ± 0.37 mJ·cm^−2^ and 7.23 ± 0.78 mJ·cm^−2^, respectively.

## 1. Introduction

Colloidal quantum dots (CQDs) are an important class of material with excellent properties for use as laser gain media, such as a high density of states, low optical gain temperature dependence, high quantum yield, and favourable photostability [1,2]. CQDs can be forced to self-assemble using an oil-in-water emulsion method to produce spherical supraparticles (SPs), with a relatively high refractive index of n_eff_ = 1.7 [3], which demonstrate whispering gallery mode (WGM) lasing under photoexcitation [4,5,6,7]. The emulsion technique for SP synthesis is facile and scalable, and resulting SPs work as microresonators, or cavities, with favourable Q factors [3,6] and low lasing thresholds [5]. WGMs are generated within SPs when light cycles around the surface of these cavities through total internal reflection enabled by the higher refractive index of an SP relative to the surrounding environment. Self-assembled SPs can be more easily tailored for multiple functionalities, for example, multi-wavelength emission for spectral coding, while their high refractive index conferred by the dense CQD assembly has the potential for stronger light confinement and, in turn, lasing in smaller structures. WGM lasers, or more generally active WGM resonators, have also gained huge interest for use in biological sensing applications due to the evanescent tail produced by WGM modes that makes the light emission of the resonators extremely sensitive to changes in the refractive index of the surrounding medium and changes to the surface chemistry [8]. Using WGM lasers for sensing is also favoured over fluorescent sensors as a result of the high signal-to-noise ratio and the reduction in emission linewidth when operating in the stimulated emission regime [9].

However, to produce functional SP laser biosensors that are water soluble and able to detect specific analytes in solution, the surface functionalisation of SPs is necessary. To our knowledge, there are currently no methods of developing functional, water soluble SP laser biosensors that can detect specific analytes in solution. As a result of the colloidal synthesis of the nanocrystals, CQDs are typically coated with hydrophobic ligands, such as oleic acid (OA), making CQDs insoluble in water and therefore unsuitable for biological applications [10]. Even though engineering of the surface chemistry of CQDs is established and has been studied extensively [7,11,12,13,14,15,16,17,18], ligand exchange directly on SPs has only recently been demonstrated [7,19,20].

Aptamers are engineered sections of DNA or RNA that can bind to specific proteins with high affinity and selectivity. For sensing applications, aptamers are ideal due to their small size, ease of production stability, and ease of modification [21]. Quantum dot aptasensors are a popular area of research, and many different types of sensors have been investigated [22,23,24,25,26,27,28,29,30]. WGM resonators have also been functionalised with aptamers and demonstrated very good sensitivity; however, these are often passive resonators made of silica or glass with stringent alignment tolerances for light injection into WGMs [22]. Thrombin is a popular target for many aptamer-functionalised sensors as it is involved in many pathological diseases, such as thrombosis and atherosclerosis. It functions as a coagulation factor and its presence in blood can indicate the presence of a blood clot. Current methods of thrombin detection are limited by the availability of suitable antibodies and can be time consuming [21,22]. 

This work addresses the above challenge by focusing on the surface functionalisation of such SPs. We report the synthesis and surface modification of SPs through the ligand exchange of oleic acid with mercaptopropionic acid (MPA) and subsequent EDC/NHS coupling with the thrombin binding aptamer TBA-15 to produce SP microlasers which could function as biosensors for the protein thrombin.

## 2. Results and Discussion

### 2.1. Synthesis and Characterisation

The synthesis of microsphere SPs consisting entirely of CdSSe/ZnS alloyed core-shell quantum dots was achieved using an oil-in-water emulsion method adapted from the literature [3]. Chloroform was chosen for its relatively low boiling point to allow the solvent to evaporate in a reasonable time at ambient temperatures, which has been shown to be the optimum drying condition for SP fabrication [31]. Environmental conditions are therefore important factors to consider during the fabrication of these microspheres and could cause large inconsistencies in the time required to produce solid SPs, which demanded constant monitoring of the emulsion to determine when the self-assembly process was finished. Surfactant choice is also important for producing SPs with favourable properties and stability [32], which is crucial for being able to alter the surface chemistry of SPs post-fabrication. The literature procedure was simplified by removing the microfluidic chip and generating emulsion droplets through vortexing the solution of CQDs and surfactant. This simplification results in broader size distributions in microspheres; however, there is still some control over the sizes of resulting SPs by manipulating the concentration of the CQD solution and surfactant used in the self-assembly process [33].

Figure 1 depicts the surface functionalisation procedure conducted in this work. The CQDs begin coated in oleic acid, a molecule with a long and unreactive carbon chain that makes the nanocrystals insoluble in water. Although this hydrophobicity is required for the SP self-assembly method, retaining this surface chemistry is not suitable for biological and sensing applications due to the inert nature of long carbon chains. A direct ligand exchange was then carried out to replace the oleic acid on the surface of the SPs with mercaptopropionic acid (MPA). This exchange is possible due to the smaller, more thermodynamically favourable chain length of MPA compared with oleic acid [13]. The resulting SPs should have carboxylic acid groups at their outer surface [12], making the SPs water soluble and able to be functionalised using a wide range of reactions. Ligand exchange was confirmed by zeta potential measurements showing a change in surface charge from −19.7 ± 6.73 mV to −31.7± 5.12 mV, as shown in Table 1, before and after ligand exchange respectively. This decrease in charge was caused by the deprotonation of carboxylic acid groups at the SP surface in an acidic environment, resulting in a more negative zeta potential value at the SP surface [34,35]. Having the carboxylic acid functionality on the outer surface of the SPs allowed for further modification of the surface. EDC/NHS coupling was subsequently used to attach the aptamer TBA-15 to the SPs through the carboxylates of the SP surface and the amine modifier attached to the 5′ end of the TBA-15 DNA chain [36]. TBA-15 was the aptamer of choice as it has the shortest DNA chain and is readily available. Consideration of the size of the molecules used for functionalisation is important because the addition of a target analyte must be within the evanescent field of the WGMs generated in the SP to be able to produce a change in lasing wavelength [37]. Functionalisation with TBA-15 was also verified using zeta potential measurements and Fourier transform infrared spectroscopy (FTIR). Zeta potential measurements before and after EDC/NHS coupling demonstrated a significant reduction in SP surface charge from −31.7 ± 5.12 mV to −12.5 ± 4.43 mV, as depicted in Table 1, which suggests successful coupling of the TBA-15 to the MPA-SP surface [38,39]. The FTIR spectra taken after each functionalisation step are shown in Figure 1, with the insets showing the areas of interest when analysing the spectra. In the spectrum for oleic acid-capped SPs (OA-SPs), the broad peak at 3120 cm^−1^ and the sharp, strong peak at 1545 cm^−1^ correspond to O–H stretching and C=O stretching for bound carboxyl groups, respectively, proving the presence of the carboxylic acid group of oleic acid [40]. The peaks at 2935 and 2865 cm^−1^ demonstrate the presence of C–H stretching from the alkene and alkane groups present in oleic acid, and peaks at 1463 and 1410 cm^−1^ also correspond to either alkane C–H bending from methyl end groups or O–H bending from the carboxylic acid and the CH_2_ groups of the carbon chain, respectively. After ligand exchange with MPA, the 18-carbon chain of oleic acid was switched with a 3-carbon chain with a thiol at one end that binds to the surface of the SPs and a carboxylic acid present at the opposite end of the chain. This replacement is reflected in the MPA-SP FTIR spectrum with the almost complete disappearance of the peaks at 2935, 2865, 1463, and 1410 cm^−1^ [41]. The FTIR spectrum taken after EDC/NHS coupling contains a peak at 3145 cm^−1^ corresponding to N–H stretching, peaks at 1637 and 1560 cm^−1^ corresponding to C=O and C–N stretching (i.e., the amide I and II absorption bands), respectively, and the peaks at 1318 and 1285 cm^−1^ correspond to the amide III band. These three indicators suggest the presence of an amide group and therefore the successful EDC/NHS coupling of TBA-15 to the surface of the SPs [42,43,44]. TBA-15 is composed of a DNA sequence consisting only of guanine and thymine bases which contain N–H bonds, possibly contributing to the strong and broad nature of the peak at 3145 cm^−1^. Alkane C–H bending peaks at 1407 and 1450 cm^−1^ are more pronounced than those in the spectrum for the MPA-capped SPs which would be expected due to an increase in the number of those bonds present after the addition of TBA-15 to the SP surface.

Comparing SP sizes at each synthetic step can afford insights into the stability of these SPs to further functionalisation. After self-assembly, the average diameter of the SPs was found to be 7.3 ± 5.3 µm, then 7.2 ± 5.1 µm after ligand exchange, and finally 2.9 ± 1.2 µm after surface functionalisation. This indicates that there is no significant impact on the average SP size after ligand exchange whereas EDC/NHS coupling has a significant reducing effect on SP size. Although the average size values before and after ligand exchange would seem to indicate that the SPs are stable for further functionalisation procedures, in fact, there is an increase in the number of malformed, damaged, or collapsed SPs that can be observed under an optical microscope after ligand exchange. The SEM images shown in Figure 2 highlight the nature of damage that could be seen, particularly after EDC/NHS coupling. To quantify this, the percentage of damaged and collapsed SPs within each sample was estimated using images taken with a camera attached to an optical microscope. For the OA-SPs, the percentage of damaged spheres was estimated to be 26%, which increased to 37% after ligand exchange and then further increased to 52% for the TBA-capped SPs (TBA-SPs). These numbers demonstrate that these SPs are only partially stable to multistep post-assembly modification and functionalisation procedures; therefore, streamlining the post-assembly processing required for other applications is essential. The conditions for the optimal EDC/NHS coupling reaction rate ideally require a buffer solution and a pH of around 8.5 [36], though it has also been found that coupling can proceed at pH 7 at a slower rate [45]. Although the OA-SPs were stable in different solvents and buffer solutions, the MPA-SPs were not stable in buffer solutions and would dissolve back into quantum dots. The MPA-SPs would also dissolve at pH 8.5, hence the coupling being carried out between pH 6.5 and 7 in water with the addition of 1–2 µL of base every 20 min to keep the pH stable. The stability of SPs when exposed to further procedures could be improved using a surface coating to add an extra barrier between the SPs and the surrounding environment, making it harder for the spheres to break down [46]. For example, encasing SPs in materials such as silica in a sol–gel process resulted in WGM lasers that were able to exhibit lasing at temperatures as high as 450 K for time periods as long as 40 min [47]. Other possibilities include the modification of SPs with short polyethylene glycol (PEG)-containing functional groups that can be used to achieve further functionalisation using techniques such as click chemistry [15,16]. 

### 2.2. Optical Measurements

The laser characteristics of the SPs were obtained by optically pumping the individual SPs using a 355 nm Nd:YAG laser, described in the Section 3. Each SP sample was diluted in water at a ratio of 1:50 SPs to water and subsequently drop cast on a glass slide which was attached to a translation stage within the setup to enable pumping of individual SPs. Figure 3 shows examples of the emission spectra and the laser transfer functions used to determine the threshold energy for individual OA-, MPA-, and TBA-SPs with diameters of 5.5 µm, 4.1 µm, and 4.5 µm, respectively. The SPs in this work demonstrate WGM lasing which is made possible by the high refractive index of the CQDs used to assemble the cavities [48]. For the laser transfer functions, Figure 3a, the spectral emission intensity at different pump energy values was integrated over the spectral range of the dominant lasing peak, for example, between 631 nm 635 nm for the OA-capped SP, 634 nm to 638 nm for the MPA-SP, and 627 nm to 633 nm for the TBA-SP. The lasing threshold was 4.1 ± 0.37 mJ·cm^−2^, 4.6 ± 0.41 mJ·cm^−2^, and 7.2 ± 0.78 mJ·cm^−2^ for the OA-SP, MPA-SP and TBA-SP, respectively. There is therefore no significant difference in the laser properties of the OA-SP and the MPA-SP, although the laser slope efficiency is slightly lower for the MPA-SP. The laser spectra are also similar with a dominating peak and a less intense, red-shifted peak attributed to the next orbital WGM order (Figure 3a,b). There is a significant increase in the threshold post surface functionalisation from 4.6 ± 0.41 mJ·cm^−2^ to 7.2 ± 0.78 mJ·cm^−2^ for the MPA-SP and TBA-SP. Such increase in threshold may be explained by the addition of molecules on the SP surface decreasing the refractive index contrast, hence the light confinement, and possibly increasing the surface roughness which in turn can affect the ability of the SP to lase. Any further molecules being attached to the surface could further affect the SP threshold which could be exploited for biosensing applications. The Q factor of the SPs measured below the threshold were similar, between 210 and 240, for OA-SPs, MPA-SPs, and TBA-SPs. Therefore, it is difficult to ascertain any effects on the Q factor from this study. The measurement above, while typical, represents one SP selected within a population and there are statistical variations sample to sample. 

To gain a better understanding on the factors affecting lasing threshold of the SPs and to establish the effect that surface functionalisation has on their lasing threshold, a one-way ANOVA (analysis of variance) was carried out using Wolfram Mathematica (Figure 4a). The ANOVA produced a high F ratio of 7.18 and a low *p*-value of 2.8 × 10^−3^ (Table 2). The F ratio is the ratio between the mean square values shown in Table 2, where a value above 1 demonstrates a difference in the means of the groups. The *p*-value tests the variance of each group, where a small *p*-value means that the standard deviations of the groups are different. Therefore, the combination of an F ratio much larger than 1 and the small *p*-value shown in Table 2 confirms that the mean threshold values of the SP groups are different. To elucidate which groups of SPs have a statistically significant difference in mean threshold values, a post hoc test (Tukey’s test) was conducted. This test found that the mean value of the threshold was significantly different between only OA- and MPA-SPs at the 5% level. These results suggest that a ligand exchange from OA to MPA increases the lasing threshold on the SPs. No other statistically significant difference was found between the samples obtained in this study. The difference in sample sizes is due to the number of SPs per 10 µL sample capable of lasing after each stage of the functionalisation process. The size distribution of SPs in Figure 5a is shown on Figure 5b. While the OA-SPs were capable of lasing for a wide range of sizes, the data suggest that the subsequent ligand exchange and EDC/NHS coupling narrow the lasing capabilities of the SPs to smaller sizes. After synthesis, the average size of the SPs before and after ligand exchange with MPA remains similar (7.3 ± 5.3 µm and 7.2 ± 5.1 µm, respectively); therefore, a reduction in the sizes of the SPs available does not explain this observed reduction in the diameter of the MPA-SPs that demonstrate lasing. However, the average size of the SPs available would be a larger factor in the observed reduction in the number of TBA-SPs that can lase due to the significant reduction in the average size of the TBA-SPs to 2.9 ± 1.2 µm. 

The optical measurements depicted in Figure 5 were taken for SPs that had been stored in water at 4 °C for 35 days to gauge their stability over time. The samples used for the initial measurements where the SPs had been drop cast onto glass slides and stored dry were not able to lase compared to the SPs that had been stored in water. After this period, the threshold was 11.1 ± 1.0 mJ·cm^−2^ for an OA-capped sphere with a diameter of 5.8 µm whereas the TBA-SP measured after this storage period had a threshold of 5.63 ± 0.50 mJ·cm^−2^ with a diameter of 3.6 µm. Here, the emission intensity was integrated over 631 and 632.5 nm for the OA-SP and 630 to 634 nm for the TBA-SP. The comparison of these thresholds measured over a range of different SPs indicates that the thresholds measured after one month of storage are comparable to the values for fresh SPs, therefore showing that the SPs are stable after storage for a month at 4 °C. Owing to the sensitivity of WGM lasing to changes in the surrounding environment and the cavity, there are several factors which could contribute to the improved threshold observed for TBA-SPs after one month of storage. Therefore, this result requires more investigation in the future [49,50].

## 3. Materials and Methods

### 3.1. Materials

CdSe_1−x_S_x_/ZnS alloyed core/shell colloidal quantum dots were purchased from CD Bioparticles and the peak emission wavelength was 630 nm. Poly(vinyl alcohol), 3-mercaptopropionic acid (MPA), *N*-(3-Dimethylaminopropyl)-*N*′-ethyl carbodiimide (EDC), and *N*-Hydroxysuccinimide (NHS) were purchased from Merck. Thrombin binding aptamer with 15 bases (TBA-15) was purchased from Integrated DNA Technologies with the sequence 5′-NH-GGT TGG TGT GGT TGG -3′ where NH is a 6-carbon amino linker.

### 3.2. Synthesis of CdSe_1−x_S_x_/ZnS Supraparticles

The procedure was adapted from the literature [3,5] and carried out under ambient conditions. A 20 mg/mL solution of CdSe_1−x_S_x_/ZnS alloyed core/shell CQDs was prepared in chloroform. Approximately 100 µL of this solution was added to 450 µL of a 2.5 w% solution of poly(vinyl alcohol) (PVA) in deionised water and stirred vigorously at room temperature for 4.5 h with constant monitoring of the emulsion after the first 2 h. The reaction mixture was centrifuged at 8000 rpm for 10 min, the supernatant was discarded, and the pellet was resuspended in DI water and stored at 4 °C. Microsphere size distributions were measured using ImageJ software using images taken from an optical microscope equipped with a Thorlabs™ camera. SEM images were captured using a JEOL JSM-IT100 operated at 20 kV. 

### 3.3. Surface Functionalisation of SPs

The procedure was adapted from literature to make it suitable for SPs rather than CQDs [35,51]. Approximately 200 µL of oleic acid-capped SPs (OA-SPs) were re-dispersed in 500 µL of a 3:1 water/ethanol mix. Excess MPA (500µL) was added, and the solution was stirred at room temperature overnight. The reaction mixture was centrifuged at 8000 rpm for 10 min and the precipitated MPA-SPs were washed with ethanol twice to remove unbound MPA. The MPA-SPs were air dried and then stored at 4 °C. 

The procedure adapted from the literature [52,53]. Approximately 200 µL of MPA-SPs were activated by the addition of 30 µL of 40 mM EDC and 30 µL of 15 mM NHS with NaOH (10 mM in H_2_O) added to increase the pH to between 7.5 and 8 before leaving the mixture to stir for 1 h. Then, 20 µL of 10 µM TBA-15 was added and the solution was left stirring for 4 h with the pH of the reaction mixture kept at approximately pH 6.5–7.0 by the addition of 2 µL of 10 mM NaOH every 20 min. The SPs were characterised using a Malvern Zeta Potentiometer and FTIR which was obtained using a Nicolet iS5 FTIR Spectrometer.

### 3.4. Optical Characterisation

Approximately 10 µL samples of SPs after each functionalisation step were drop cast on glass slides and left to dry. Lasing measurements of self-assembled microlasers were obtained using a 355 nm, 5 ns pulsed Nd:YAG laser at a 10 Hz repetition rate with a beam spot area of ~2.6 × 10^−5^ cm^2^. The beam intensity was altered using a variable wheel neutral density attenuator and focused on the sample using a 10× magnified objective lens. An Avantes AvaSpec-2048-4-DT spectrometer was used to acquire spectral data. 

## 4. Conclusions

In conclusion, SPs of CdSeS/ZnS CQDs were successfully modified with the thrombin binding aptamer TBA-15 and are capable of lasing post-functionalisation, with lasing functionality retained after storage in water for over 1 month. Surface functionalisation is essential to realising the potential of these self-assembled SP lasers for a myriad of applications; therefore, this proof of concept paves the way for the use of SPs in biosensing applications. While surface functionalisation could reduce the toxicity of SPs, due to the nature of toxicity of the CQDs used, these SPs would only be used for in vitro sensing, for example, in a lab-on-a-chip sensing device. Using a facile and relatively mild procedure, the oleic acid originally coating the surface of the SPs can be exchanged with MPA to create a carboxylic acid-coated surface which can undergo EDC/NHS coupling to attach TBA-15 to the surface of the SPs in a three-step modification process. Zeta potential values of −21.5 ± 7.22 mV, +24.1 ± 4.99 mV, and −22.4 ± 7.13 mV for OA-, MPA-, and TBA-SPs, respectively, confirmed the successful surface functionalisation, along with the FTIR spectra. The SPs before and after functionalisation demonstrated low lasing thresholds, with an OA- and TBA-SP exhibiting thresholds as low as 4.10 ± 0.37 mJ·cm^−2^ and 7.23 ± 0.78 mJ·cm^−2^, respectively. The thresholds of an OA- and TBA-SP were measured after storage in water for over 1 month and remained similar to those measured for fresh SPs, with values of 11.1 ± 1.0 mJ·cm^−2^ and 5.63 ± 0.50 mJ·cm^−2^ for the OA- and TBA-SP, respectively. Obtaining SPs with this surface chemistry creates a platform that is incredibly versatile because EDC/NHS coupling is a very well-known reaction that can be used to attach any molecule containing an amine group to the surface of MPA-capped SPs [36]. The stability of SPs when exposed to further procedures could be improved using a surface coating to add an extra barrier between the SPs and the surrounding environment, making it harder for the spheres to break down [46]. For example, encasing SPs in materials such as silica enabled lasing at temperatures as high as 450 K for time periods as long as 40 min [47]. Other possibilities include the modification of SPs with short polyethylene glycol (PEG) chains containing functional groups that can be used to achieve further functionalisation using different techniques [15,16]. Adding an extra layer of protection to the SPs could also improve the stability of the spheres over time; however, these SPs have already been shown to be stable when stored in water. Work is currently underway to investigate the biosensing capabilities of these SPs.

## Data Availability

The dataset can be found at: https://doi.org/10.15129/d8b31125-23cb-4325-b136-38073dd6a4a6, 18 September 2023.

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
