# Peer review of "Surface Functionalisation of Self-Assembled Quantum Dot Microlasers with a DNA Aptamer"

_ijms, 2023, doi:10.3390/ijms241914416_

Round 1

Reviewer 1 Report

This manuscript presents interesting study on forming spherical supraparticles (SPs) from semiconductor quantum dots, exchanging the ligand on the surface of SPs and modifying the SPs surface by coupling reaction. The authors demonstrated a very recent approach in the literature to preform ligand exchange study for SPs instead of the well-studied monodisperse nanocrystals, and applied their SPs for microlasers, potentially biosensor. Therefore, I suggested this manuscript could be published in International Journal of Molecular Sciences. My further comments are listed below.

1. The authors prepared SPs by adding commercially available colloidal quantum dots chloroform solution into poly(vinyl alcohol) (PVA) aqueous solution. Could the hydroxy groups in PVA partially bond to SPs and replace oleic acid ligands? Which will be helpful for the SPs resuspended in DI water or 3:1 water/ethanol mixture?

2. For characterization, the SEM images only showed single micro size particle. Large-scale SEM images containing multiple particles are suggested, so the readers can know the morphologies and size distribution of SPs. Also, for a single SP, high resolution SEM or TEM may be interesting to observe the particle surface morphologies for the three different SPs.

3. Chemical information of MPA should be added in section 3.1 Materials. There are some other characters appeared at the center of figure 5.

Author Response

Response to Comment 1: Thank you for this really interesting question. While PVA would certainly interact with the oleic acid ligands bound to the quantum dots within SPs and the QDs themselves, we believe that replacing oleic acid with the longer PVA chains would not be entropically favourable. Additionally, the PVA solution used for the SP assembly is probably not concentrated enough to make this exchange possible on a wide enough scale in the samples.

Response to Comment 2: We agree with this. We have reworked Figure 2 to add SEM images of the whole samples and added close up images of a particle surface where possible. (Page 5 and removed the following comment from the text: “and in generally larger OA-SPs and MPA-SPs”).

Response to Comment 3: Thank you for pointing this out. We have added the MPA information to the list of materials purchased in section 3.1 and fixed the problem with Figure 5. (Pages 8 and 9).

Reviewer 2 Report

Quantum dots and their application for the delivery of medicinal preparations and theranostics, this is one of the trending topics of modern science about meterials.

The goal of the authors of the work, well stated and understandable, is to create a genetic vector for the delivery of a quantum dot, for subsequent light therapy or imaging/diagnostics.

In this regard, I have a number of fundamental questions.

The choice of CDSes/ZnS quantum dots is very logical and convenient from the point of view of science and interpretation of the results, but for the purposes of theranostics they are not applicable due to toxicity. Toxicity can be reduced by their surface modification.

However, the authors did not evaluate the toxicity of the final conjugate of the quantum dot and the genetic vector.

The second question is the wavelength of the excitation and emission of the quantum dot.

The work indicates that the excitation wavelength is 355nm, this is a fairly harsh UV irradiation, and it does not penetrate deep into the human body. For the purposes of diagnosis or therapy, it is necessary to use the excitation near the "transparency windows".

The same applies to emissions.

Technical issues.

Why the measurement of the zeta potential was carried out at different pH, for comparison, you need to clean the quantum dots in the ass of each modification and conduct research in the buffer.

Figure 1. For easier perception of graphs, I recommend putting a legend on the drawing.

Figure 2. The SEM image is not representative. There is only one particle in the frame, a snapshot of a large field is needed (for statistical reliability) and the particle size distribution.

Author Response

Response to Comment 1: We agree, these SP lasers would be unsuitable for in vivo biosensing. We believe these lasers should be used in a lab on a chip configuration for in vitro sensing. We have altered the conclusion to make this point clearer. (Page 9, Paragraph 4, Lines 281-283).

Response to Comment 2: This is another important point to consider when creating biosensors. As a result of the toxicity of the QDs used in this study, these SPs would be used for in vitro biosensing in a lab on a chip device or a benchtop design. There are possibilities with these CQDs to pump at a longer and less harsh wavelengths, for example visible light at a wavelength above the bandgap of the CQDs. The alteration made is addressed in the above response.

Response to Comment 3: We agree with this comment and have reworked Table 1 with the new zeta potential results carried out at similar pH and changed the text accordingly (Page 3, Paragraph 2, Lines 99 – 102 and Line 113). Unfortunately, while it would be preferable for this work to be carried out in buffer, it tends to cause the complete dissolution of MPA- and TBA-SPs back into quantum dots. We have tried dissolving in buffer solutions using more bulky salts, however this results in the same dissolution.

Response to Comment 4: Thank you for pointing this out. We have added a legend in Figure 1 and reworked Figure 2 to add SEM images of whole samples and added a size distribution of the SPs measured by optical microscopy.

Round 2

Reviewer 2 Report

All comments have been taken into consideration.